

# Hyaluronic acid plasma levels during high *versus* low tidal volume ventilation in a porcine sepsis model

Rainer Thomas[1], Tanghua Liu[1], Arno Schad[2], Robert Ruemmler[1],
Jens Kamuf[1], René Rissel[1], Thomas Ott[1], Matthias David[1],
Erik K. Hartmann[1] and Alexander Ziebart[1]

[1] Department of Anesthesiology, Medical Centre of the Johannes Gutenberg University, Mainz, Germany
[2] Institute of Pathology, Medical Center of the Johannes Gutenberg-University, Mainz, Germany

## ABSTRACT

**Background:** Shedding of the endothelial glycocalyx can be observed regularly during sepsis. Moreover, sepsis may be associated with acute respiratory distress syndrome (ARDS), which requires lung protective ventilation with the two cornerstones of application of low tidal volume and positive end-expiratory pressure. This study investigated the effect of a lung protective ventilation on the integrity of the endothelial glycocalyx in comparison to a high tidal volume ventilation mode in a porcine model of sepsis-induced ARDS.

**Methods:** After approval by the State and Institutional Animal Care Committee, 20 male pigs were anesthetized and received a continuous infusion of lipopolysaccharide to induce septic shock. The animals were randomly assigned to either low tidal volume ventilation, high tidal volume ventilation, or no-LPS-group groups and observed for 6 h. In addition to the gas exchange parameters and hematologic analyses, the serum hyaluronic acid concentrations were determined from central venous blood and from pre- and postpulmonary and pre- and postcerebral circulation. Post-mortem analysis included histopathological evaluation and determination of the pulmonary and cerebral wet-to-dry ratios.

**Results:** Both sepsis groups developed ARDS within 6 h of the experiment and showed significantly increased serum levels of hyaluronic acid in comparison to the no-LPS-group. No significant differences in the hyaluronic acid concentrations were detected before and after pulmonary and cerebral circulation. There was also no significant difference in the serum hyaluronic acid concentrations between the two sepsis groups. Post-mortem analysis showed no significant difference between the two sepsis groups.

**Conclusion:** In a porcine model of septic shock and ARDS, the serum hyaluronic acid levels were significantly elevated in both sepsis groups in comparison to the no-LPS-group. Intergroup comparison between lung protective ventilated and high tidal ventilated animals revealed no significant differences in the serum hyaluronic acid levels.

Corresponding author
Alexander Ziebart,
alexander.ziebart@unimedizin-mainz.de

## INTRODUCTION

The endothelial glycocalyx (EG) is a thin gel-like layer composed of proteoglycans and glycoproteins found on the luminal surface of healthy blood vessels (*Ruane-O'Hora, Ahmeda & Markos, 2020*; *Uchimido, Schmidt & Shapiro, 2019*). First described in the 1960s, many physiological interactions between the glycocalyx and endothelial cells have been discovered which influence physiological and pathological processes. It has been postulated that the EG plays an important role as part of the vascular barrier (*Sieve, Münster-Kühnel & Hilfiker-Kleiner, 2018*), in mechanotransduction (*Tarbell & Pahakis, 2006*), in the development of diabetes mellitus (*Dogné, Flamion & Caron, 2018*) and atherosclerosis (*Sieve, Münster-Kühnel & Hilfiker-Kleiner, 2018*), in ischemia/reperfusion injury (*Dogné & Flamion, 2020*), as well as in inflammation and sepsis (*Dogné & Flamion, 2020*; *Uchimido, Schmidt & Shapiro, 2019*).

In the last years, several experimental and clinical studies have shown that sepsis damages the EG. This results in endothelial dysfunction and may initiate the development of edema. The resulting impaired lung function eventually leads to the development of acute respiratory distress syndrome (ARDS) (*Huang et al., 2018*; *LaRivière & Schmidt, 2018*). The same applies to the brain: Shedding of the EG leads to an increased capillary permeability resulting in edema (*Jin et al., 2021*; *Zhu et al., 2018*). In these studies, EG degradation was measured indirectly by increased plasma concentrations of EG components, such as heparan sulfate, syndecan-1, or hyaluronic acid (HA).

One of the cornerstones in the management of ARDS during sepsis is protective mechanical ventilation with low tidal volumes ($V_t$ 6 ml $kg^{-1}$ body weight), which reduces mortality and barotrauma and provides a higher weaning rate than conventional ventilation strategies (*Maccagnan Pinheiro Besen, Tomazini & Pontes Azevedo, 2021*).

Mechanical ventilation itself can induce an inflammatory response, thus inducing shedding of the EG as it is happening during sepsis. Several studies describe effects of different ventilation strategies on systemic inflammation. It has been demonstrated that a protective ventilation strategy during major surgery leads to an attenuated inflammatory response compared with non-protective ventilation (*Michelet et al., 2006*; *Wrigge et al., 2004*). It has been shown, that protective ventilation in ARDS reduces cytokine response compared to ventilation with higher tidal volumes and less PEEP (*Ranieri et al., 1999*).

Despite these findings, it remains unknown whether different ventilatory strategies influence the shedding of the EG by reducing or aggravating an inflammatory response. Therefore, we proposed two hypotheses:

– Sepsis leads to depletion of EG as measured by plasma HA levels.
– The selected $V_t$ influences the extent of pulmonary but not cerebral EG depletion, measured by increased plasma levels in the pulmonary circulation, but not in the cerebral circulation.

## METHODS

### Animals

The prospective randomized animal experiment was approved by the State and Institutional Animal Care Commission (Landesuntersuchungsamt Rheinland-Pfalz, Koblenz, Germany; reference number: G12-1-059) in accordance with ARRIVE guidelines and planed and conducted in the laboratories of the Department of Anesthesiology of the Medical Centre of the Johannes Gutenberg-University Mainz. Twenty-six German landrace pigs (weight, 24–27 kg; age, 8–12 weeks) were bred from a local farmer, who was recommended by the State and Institutional Animal Care Committee. The group size was based on the trials of previous large animal studies.

To minimize stress, the animals remained in their familiar surroundings under breeder-controlled environmental conditions as long as possible. Food, but not water, was withheld 6 hours before the scheduled experiment to reduce the risk of aspiration. Shortly before the start of the experiment, the animals were delivered from the breeder under controlled and gentle conditions to the state-certified and approved members of the laboratory until the end of the experiments.

### Anesthesia and animal preparation

Sedation of the pigs was achieved by intramuscular injection of midazolam ($0.2$ mg kg$^{-1}$) and ketamine ($8$ mg kg$^{-1}$). Anesthesia was induced by intravenous injection of fentanyl ($10$ μg kg$^{-1}$) and propofol ($4$ mg kg$^{-1}$) through an ear-vein cannula, followed by administration of atracurium ($1$ mg kg$^{-1}$) to induce muscular paralysis and facilitate endotracheal intubation. Anesthesia was maintained using a propofol ($8$–$12$ mg kg$^{-1}$ h$^{-1}$) and fentanyl ($6$ μg kg$^{-1}$ h$^{-1}$) infusion. The animals were ventilated in a volume-controlled mode (AVEA, CareFusion, San Diego, CA, USA).

First, a central venous line was placed in one femoral vein, followed by the placement of a PiCCO-catheter (Pulsion, Munich, Germany) for extended hemodynamic monitoring *via* femoral access. A left ventricular catheter and a pulmonary artery catheter were placed in the other femoral vein and artery *via* introducers. Furthermore, the right carotid artery was catheterized with an antegrade pressure line and the right jugular vein with a retrograde catheter. All the catheters were placed under ultrasound guidance using the Seldinger technique.

The hemodynamic and spirometric parameters were recorded continuously with the Datex S/5 device (Datex-Ohmeda GmbH, Duisburg, Germany) and the PiCCO device.

### Experimental protocol

Only animals that achieved ARDS criteria were included in the intervention groups (*Tiba et al., 2021*).

After instrumentation and a brief stabilization period of 15 min, baseline measurements were performed. The animals were then randomly divided into three groups:

(1) LPS administration with low tidal volume ventilation with (LTV) ($7$ ml kg$^{-1}$; $n = 7$)

(2) LPS administration with high tidal volume ventilation (HTV) ($15$ ml kg$^{-1}$; $n = 7$)

(3) No LPS administration, with low tidal volume ventilation (no LPS; $n = 6$)

In groups (1) and (2) systemic septic shock-like conditions were simulated by intravenous administration of lipopolysaccharides (LPS; Echerichia coli O111:B4, Sigma-Aldrich, St. Louis, MO, USA) starting with 160 µg kg$^{-1}$ h$^{-1}$ for 30 min and continuous administration of 16 µg kg$^{-1}$ h$^{-1}$ for the remainder of the experiment. Premature euthanasia of the animals was foreseen if animals did not achieve ARDS criteria, unpredictable stress and pain were observed or if uncontrollable complications such as excessive bleeding or intubation problems occurred during the preparation. Six animals did not achieve ARDS and were euthanized like described below. Major complications never occurred.

All the animals were ventilated with a positive end-expiratory pressure (PEEP) of 5 mbar, an inspired oxygen fraction of 0.4, and a variable respiratory rate to maintain normocapnia. When peripheral oxygen saturation fell below 93% for more than 5 min, the ventilator settings were changed according to a defined schedule (Table 1). The no LPS group was observed for 6 h and received the same ventilation as the LTV group, but with no administration of LPS (Fig. 1).

For fluid replacement, Sterofundin ISO (Braun, Melsungen, Germany) was infused continuously at a rate of 10 ml kg$^{-1}$ h$^{-1}$. To avoid hemodynamic deterioration, a standardized fluid bolus (250 ml hydroxyethyl starch over 30 min; Volulyte 6% 130/0.4, Fresenius Kabi, Bad Homburg, Germany) was administered during preparation, followed by 90 ml h$^{-1}$ hydroxyethyl starch during the experiment. To ensure stable hemodynamic conditions, a continuous infusion of norepinephrine was administered during the experiment to maintain mean arterial pressure above 60 mmHg.

At the end of observation, the animals were euthanized under deep anesthesia by central venous injection of propofol (200 mg) and potassium chloride (40 mmol).

## Hyaluronic acid plasma levels and hematological parameters

Blood samples for the HA concentrations as primary outcome parameter were drawn from the central venous line, pulmonary artery, left ventricular catheter, carotid artery catheter, and jugular vein catheter at baseline, at 3 h, and at 6 h. The plasma HA levels were double determined by ELISA (Echelon, Salt Lake City, UT, USA). Heparan sulfate ELISA kits (AMSBio HS ELISA Kit; Milton, UK) were used as an additional EG degradation marker, but they failed to produce valid results in our porcine model. We also analyzed the leukocyte and platelet counts, lactate plasma levels and arterial blood gases.

## Post-mortem analysis

Lungs from the HTV and LTV groups were removed en-bloc. The right lung was sliced, weighed, and dried (60 °C for 48 h) to determine the pulmonary wet-to-dry ratio. The frontal cortex was treated in the same manner to determine the cerebral wet-to-dry ratio. The left lung was incised beginning at the main bronchus and three samples were taken from each lobe, one near the hilum, one at the end of the inferior and superior main bronchus, and one midway between the others, representing gravitational

**Table 1 Intervention scheme.**

|  | LTV | | HTV | |
|---|---|---|---|---|
| Step 1 | FiO$_2$ 0.4 | PEEP 5 cmH$_2$O | FiO$_2$ 0.4 | PEEP 5 cmH$_2$O |
| Step 2 | FiO$_2$ 0.4 | PEEP 5-8 cmH$_2$O | FiO$_2$ 0.45 | PEEP 5 cmH$_2$O |
| Step 3 | FiO$_2$ 0.4 | PEEP 8-10cmH$_2$O | FiO$_2$ 0.5 | PEEP 5 cmH$_2$O |
| Step 4 | FiO$_2$ 0.5 | PEEP 10 cmH$_2$O | FiO$_2$ 0.55 | PEEP 5 cmH$_2$O |
| Step 5 | FiO$_2$ 0.6 | PEEP 10 cmH$_2$O | FiO$_2$ 0.6 | PEEP 5 cmH$_2$O |
| Step 6 | FiO$_2$ 0.6 | PEEP10-14cmH$_2$O | FiO$_2$ 0.65 | PEEP 5 cmH$_2$O |
| Step 7 | FiO$_2$ 0.6 | PEEP 15 cmH$_2$O | FiO$_2$ 0.7 | PEEP 5 cmH$_2$O |
| Step 8 | FiO$_2$ 0.7 | PEEP 15 cmH$_2$O | FiO$_2$ 0.75 | PEEP 5 cmH$_2$O |
| Step 9 | FiO$_2$ 0.8 | PEEP 15 cmH$_2$O | FiO$_2$ 0.8 | PEEP 5 cmH$_2$O |
| Step 10 | FiO$_2$ 0.9 | PEEP 15 cmH$_2$O | FiO$_2$ 0.85 | PEEP 5 cmH$_2$O |
| Step11 | FiO$_2$ 1.0 | PEEP 15 cmH$_2$O | FiO$_2$ 0.9 | PEEP 5 cmH$_2$O |
| Step 12 | | | FiO$_2$ 0.95 | PEEP 5 cmH$_2$O |
| Step 13 | | | FiO$_2$ 1.0 | PEEP 5 cmH$_2$O |

**Note:**
If SpO$_2$ dropped below 93% for > 5 min.

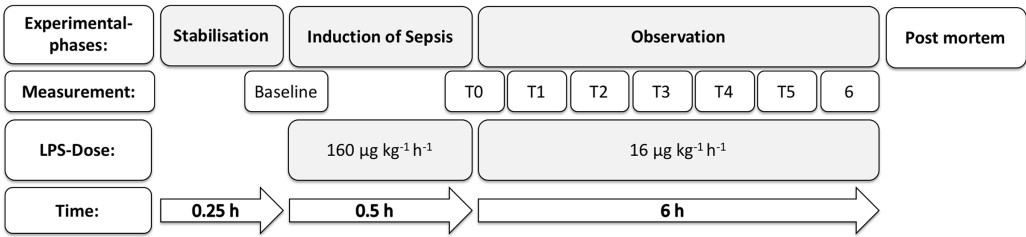

**Figure 1 Experimental flow chart.** The top line represents the experimental phases. Starting with stabilization after general anesthesia induction and preparation (stabilization), septic shock was induced by a high dose of LPS (160 µg kg$^{-1}$ h$^{-1}$) over 30 min. Then, the infusion rate of LPS was reduced to 16 µg kg$^{-1}$ h$^{-1}$. This was followed by a 6-h observation period (observation). The third line indicates the LPS dose. The fourth line indicates the duration of the periods.

dependent, central and nondependent regions. The sections were fixed in formalin, embedded in paraffin, stained with hematoxylin and eosin and analyzed under the supervision of an experienced pathologist. An established histopathologic score was used to determine tissue damage, which consisted of seven parameters (alveolar edema, interstitial edema, hemorrhage, inflammatory infiltration, epithelial destruction, microatelectasis, and overdistension). For each parameter the severity and the extent of the involvement in each slice was determined and scored for four fields of view (values from 0 to 5; 0 = normal appearance, 5 = severe effect). Additional, the quantity of each parameter was analyzed for the complete slice (0 to 5; 0 = 0% extension, 5 = 100% extension). Both scores where multiplied, resulting in a summary score with a maximum of 175 points (*Ziebart et al., 2014*; *Spieth et al., 2011*; *Spieth et al., 2007*).

## Statistical analysis

Data are presented as mean and standard deviation (SD). Box plots are presented as median and percentile. The effects of the group and intergroup differences were assessed using two-way analysis of variance (ANOVA), and paired multiple comparison correction was performed using the Student–Newman–Keuls method. For the post-mortem parameters, the Kruskal–Wallis test and post-hoc Dunn test were used. Statistical analysis was performed using SigmaPlot 12.5 software (Systat Software, San Jose, CA, USA).

All hemodynamic, spirometric, and laboratory parameters were analyzed from all animals. For simplicity, only the data points every 2 h after ARDS were shown.

## RESULTS

### Ventilation and hemodynamics

All the included animals in the LTV and HTV groups developed gas exchange dysfunction corresponding to at least mild ARDS ($PaO_2\, F_iO_2^{-1} < 300$), whereas no ARDS was observed in animals in the no LPS group. After 3 and 6 h, the ratio of arterial partial pressure of oxygen to $F_iO_2$ was significantly lower in both sepsis groups. The peak ventilator pressures were significantly higher in the LTV and HTV groups than the no LPS group. PEEP was also significantly increased in the LTV group in comparison to the HTV and no LPS groups after 3 and 6 h.

At baseline, all hemodynamic parameters showed no significant differences among the three groups. During the entire experimental period, mean arterial blood pressure (MAP) was significantly lower in the LTV and HTV groups compared to the no LPS group, but it was maintained above 60 mmHg at all-time points by the administration of norepinephrine.

A significant decrease in cardiac output was only observed in the LTV group at the 6-h time point; no significant decrease was observed in the HTV or no LPS groups. Mean pulmonary arterial (PA) pressure was significantly lower in the no LPS group in comparison to the two intervention groups after 6 h.

While no norepinephrine was required to maintain MAP above 60 mmHg in the no LPS group, dosing increased significantly in comparison to the baseline measurements in both the LTV and HTV groups, with animals in the LTV group requiring significantly higher doses than the HTV group (Table 2).

### Hyaluronic acid concentrations

Analysis of the HA concentration showed no significant difference between the three study groups at baseline. After 3 and 6 h, the HA acid concentrations were found to be significantly higher in the LTV and HTV groups compared to their baseline concentrations and the no LPS group (Fig. 2). In the no LPS group, the 6-h values were higher than the baseline concentrations, but they were significantly lower than the concentrations in the LTV and HTV groups at the same time points. No significant differences between the LTV and HTV groups were measurable before and after the brain or lung passage (Fig. 2).

**Table 2 Ventilatory, hemodynamic, and hematologic data.**

| Parameter | | Baseline | 3h | 6h |
|---|---|---|---|---|
| Mean (SD) | | | | |
| **Ventilation** | | | | |
| | no LPS | 16 ± 0.5 | 16 ± 0.3#[2] | 16 ± 0.8#[2] |
| $P_{eak}$ (cm $H_2O$) | LTV | 16 ± 1.6 | 23 ± 11.6 | 27 ± 4.5 |
| | HTV | 17 ± 3 | 31 ± 4.5#[2] | 30 ± 4.3#[2] |
| $P_{mean}$ (cm $H_2O$) | no LPS | 9 ± 0.2 | 9 ± 0.2#[1,2] | 9 ± 0.2#[1,2] |
| | LTV | 9 ± 0.8 | 11 ± 11 #[1] | 16 ± 4.4 #[1] |
| | HTV | 9 ± 1 | 13 ± 1.1#[2] | 12 ± 1.1#[2] |
| $P_{plat}$ (cm $H_2O$) | no LPS | 12 ± 1 | 13 ± 1 | 13 ± 1 |
| | LTV | 12 ± 1 | 22 ± 12#[1] | 25 ± 5#[1] |
| | HTV | 14 ± 2 | 29 ± 5#[2] | 27 ± 5#[2] |
| | no LPS | 466 ± 52 | 437 ± 53#[1,2] | 430 ± 65#[1,2] |
| $PaO_2\ Fi0_2^{-1}$ | LTV | 459 ± 71 | 200 ± 85#[1] | 210 ± 43#[1] |
| | HTV | 419 ± 57 | 278 ± 89 #[2] | 188 ± 30#[2] |
| | no LPS | 7.3 ± 0.2 | 7.2 ± 0.2#[2] | 7.3 ± 0.1#[2] |
| $V_t$ (ml $kg^{-1}$) | LTV | 7.3 ± 0.6 | 7.0 ± 0.5#[3] | 7.3 ± 0.4#[3] |
| | HTV | 7.3 ± 0.3 | 15.5 ± 0.7#[2,3] | 15.2 ± 0.5#[2,3] |
| | no LPS | 36.0 ± 2.6 | 36.2 ± 2.6#[2] | 34.7 ± 1.7#[2] |
| RR ($min^{-1}$) | LTV | 36.6 ± 6.0 | 44.1 ± 4.0#[3] | 46.3 ± 7.4#[3] |
| | HTV | 38.9 ± 3.9 | 18.6 ± 2.3#[2,3] | 17.7 ± 2.0#[2,3] |
| | no LPS | 8 ± 1 | 7 ± 1#[2] | 7 ± 1#[2] |
| PEEP (cm $H_2O$) | LTV | 8 ± 1 | 8 ± 10#[3] | 10 ± 3#[3] |
| | HTV | 8 ± 0.3 | 5 ± 1#[2,3] | 5 ± 0.4#[2,3] |
| | no LPS | 0.36 ± 0 | 0.36 ± 0#[1] | 0.36 ± 0#[1,2] |
| $F_iO_2$ | LTV | 0.36 ± 0 | 0.49 ± 0.2#[1,2] | 0.46 ± 0.1#[1] |
| | HTV | 0.36 ± 0 | 0.37 ± 0#[2] | 0.47 ± 0.2#[2] |
| **Hemodynamics** | | | | |
| | no LPS | 89 ± 11 | 91 ± 14#[1,2] | 94 ± 22#[1,2] |
| HF ($min^{-1}$) | LTV | 93 ± 15 | 206 ± 28#[1] | 193 ± 23#[1] |
| | HTV | 101 ± 16 | 147 ± 39#[2] | 181 ± 57#[2] |
| | no LPS | 95 ± 21 | 93 ± 8#[1,2] | 76 ± 3#[1,2] |
| MAP (mmHg) | LTV | 85 ± 13 | 66 ± 2#[1] | 60 ± 5#[1] |
| | HTV | 90 ± 16 | 72 ± 13#[2] | 64 ± 20#[2] |
| | no LPS | 3 ± 0.4 | 3.4 ± 0.3 | 3.1 ± 0.4 |
| CO (l $min^{-1}$) | LTV | 3.7 ± 0.8 | 3.1 ± 1.0 | 2.3 ± 1.5 |
| | HTV | 3.8 ± 0.7 | 2.9 ± 0.3 | 2.6 ± 1.8 |
| | no LPS | 22 ± 5 | 22 ± 3 | 26 ± 5 |
| PA (mmHg) | LTV | 21 ± 4 | 43 ± 12#[1] | 39 ± 13#[1] |
| | HTV | 22 ± 3 | 45 ± 13#[2] | 47 ± 17#[2] |
| | no LPS | 0 | 0#[1,2] | 0#[1,2] |

(Continued)

| Table 2 (continued) | | | | | |
|---|---|---|---|---|---|
| Parameter | | Baseline | 3h | | 6h |
| NA ($\mu$g kg$^{-1}$ min$^{-1}$) | LTV | 0 | 0.67 ± 0.65#[1] | | 25.3 ± 18.2#[1] |
| | HTV | 0 | 0.23 ± 0.3#[2] | | 4 ± 12.2#[2,3] |
| Hematologic | | | | | |
| | no LPS | 14.5 ± 5.6 | 17.2 ± 4.7#[1,2] | | 16.3 ± 4.3#[1,2] |
| Leukocytes (nl$^{-1}$) | LTV | 10 ± 3.7 | 1.3 ± 0.5#[1] | | 1.9 ± 1#[1] |
| | HTV | 14 ± 4.6 | 1.3 ± 0.2#[2] | | 1.4 ± 0.6#[2] |
| | no LPS | 225 ± 130 | 227 ± 102#[1,2] | | 218 ± 65#[1,2] |
| Thrombocytes (nl$^{-1}$) | LTV | 211 ± 68 | 149 ± 85#[1] | | 157 ± 116#[1] |
| | HTV | 313 ± 103 | 154 ± 96#[2] | | 99 ± 148#[2] |

Note:
P$_{eak}$, Peak inspiratory pressure; P$_{mean}$, mean airway pressure; P$_{plat}$, Plateau pressure; PaO$_2$ Fi0$_2^{-1}$, oxygenation ratio; V$_t$, tidal volume; RR, respiratory ratio; PEEP, positive end-expiratory pressure; FiO$_2$, fraction of inspired oxygen; HF, heart frequency, MAP, mean arterial pressure; CO, cardiac output; PA, mean arterial pressure; NA, dose of Norepinephrine; #$p$ < 005 intergroup difference: 1 = no-LPS-group *vs.* LTV, 2 = no-LPS-group *vs.* HTV, 3 = LTV *vs.* HTV.

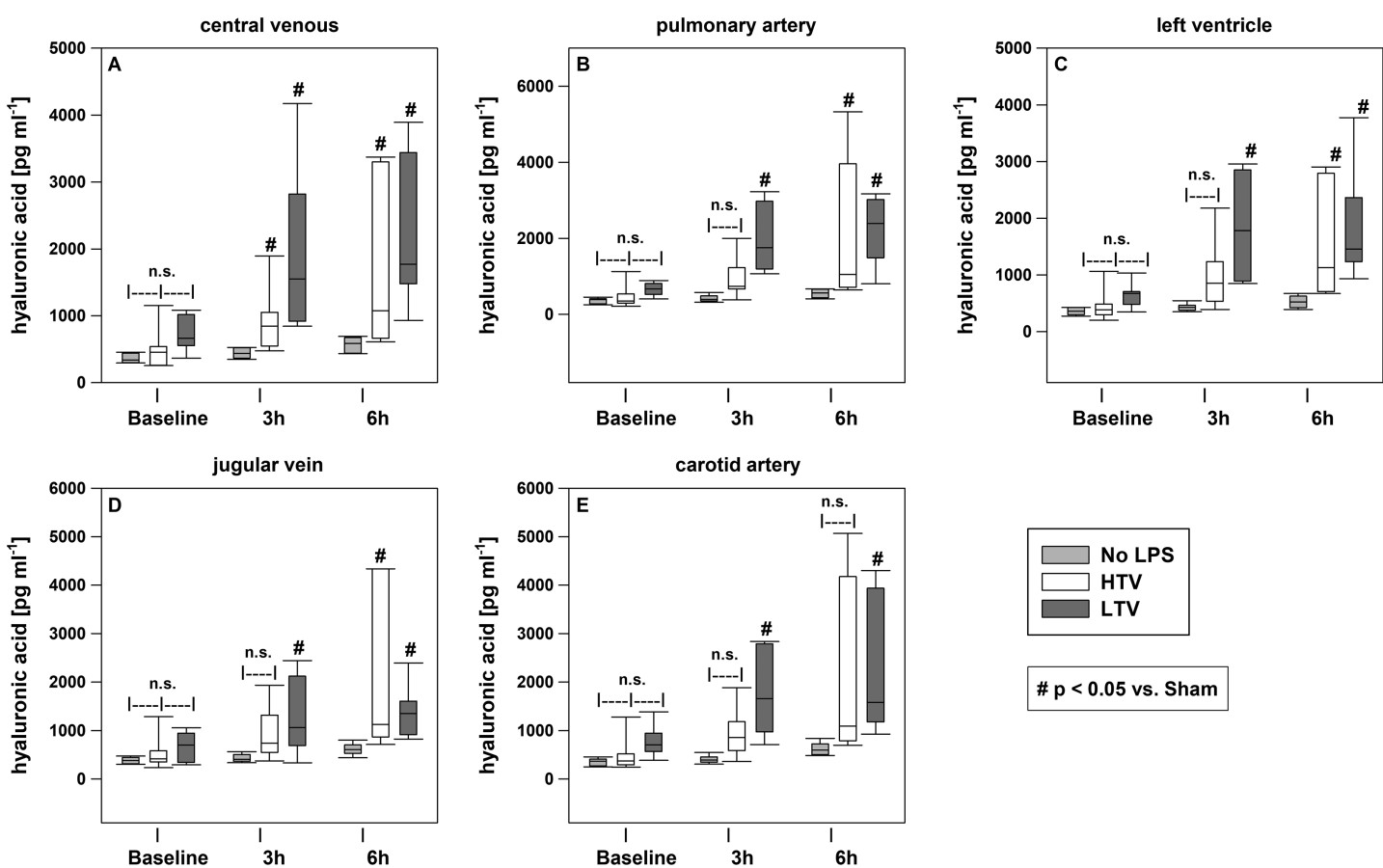

**Figure 2 Central venous HA concentrations.** HA plasma level determined by enzyme-linked immunosorbent assays at baseline, after 3 and 6 h. Kruskal–Wallis test with the post-hoc Dunn test. The line in the boxes is the median; the top and bottom of the box represent the interquartile range; data within the whiskers are in the range of 1.5 times the interquartile range. #$p$ < 0.05 *vs.* no-LPS-group; $n$ = 20.

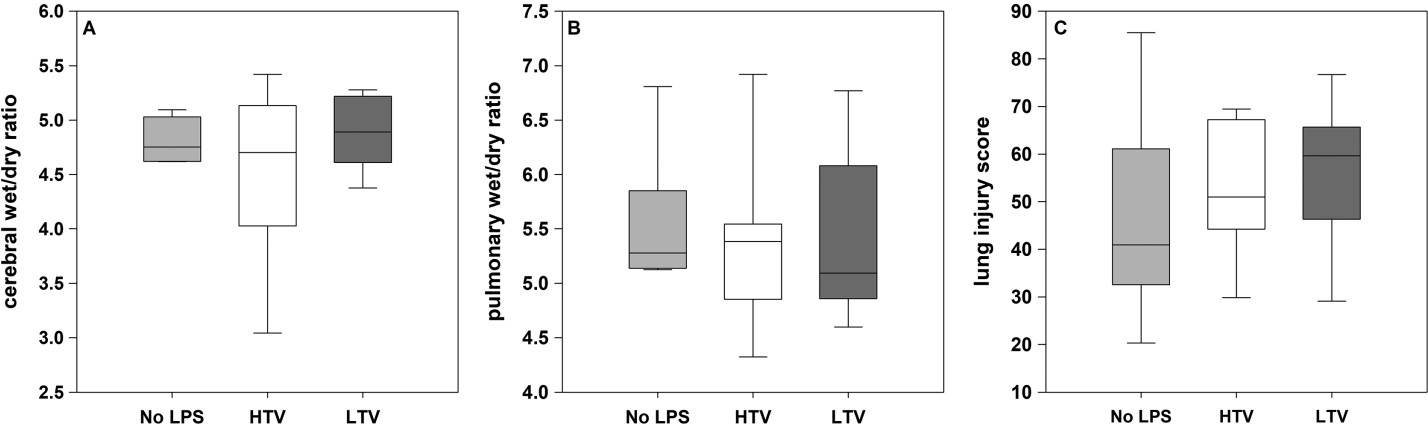

**Figure 3 Cerebral and pulmonary post-mortem analysis.** (A) Cerebral wet/dry ratio; (B) pulmonary wet/dry ratio; (C) lung injury score; Kruskal–Wallis test with the post-hoc Dunn test. The line in the boxes is the median; the top and bottom of the box represent the interquartile range; data within the whiskers are in the range of 1.5 times the interquartile range. The pulmonary and cerebral post-mortem analysis showed no significant difference between the experimental groups.

## Hematological parameters

The blood count results showed a significant decrease in leukocytes and platelets in the LTV and HTV groups compared to baseline conditions and the no LPS group over the course of the experiment. A significant increase in serum hematocrit was also observed in both sepsis groups. Moreover, the lactate levels were significantly increased in the sepsis groups in comparison to the no LPS group, with a more pronounced increase in the LTV group.

## Post-mortem assessment

The pulmonary and cerebral wet-dry ratios showed no significant difference between the experimental groups. The global scoring values and the regional distribution showed no significant difference between the three groups due to the high variances (Fig. 3).

## DISCUSSION

The present study found significantly increased plasma HA levels in the central venous, pulmonary, and cerebral circulation in a porcine model of sepsis-induced ARDS. Furthermore, the pulmonary HA release was not specifically influenced by the different tidal volumes in the short run. Therefore, the first hypothesis of the study was confirmed, but the second hypothesis was not.

All animals in the LTV and HTV groups developed ARDS according to the current definitions (*Ranieri et al., 2012*). The fact that no significant differences in the oxygenation ratio were found when comparing the two sepsis groups is surprising only at first glance. It is well known that the benefits of a protective ventilation regime become apparent only after a period of time that is longer than the 6 h, which was the duration of our experiment (*Biener et al., 2013*). In contrast, higher tidal volumes are able to recruit atelectatic lung tissue and improve gas exchange, but they represent a double-edged sword by inducing ventilator-induced lung injury in the long run. Furthermore, a previous

study demonstrated that LPS impairs hypoxic pulmonary vasoconstriction, which may lead to a greater decrease in the oxygenation rate even with a protective ventilator regime (*Davieds et al., 2016*; *Turzo et al., 2018*). HA is an established serum marker for shedding of the EG. A closer look at the EG suggests that HA is the first component of EG shedding because it is closest to the vessel lumen compared with heparan sulfates or even deeper transmembrane proteins (*Dogné & Flamion, 2020*; *Martin et al., 2016*).

Consequently, the significant increase in plasma HA levels provides evidence of damage to the EG in both sepsis groups. Previous studies confirm that inflammatory processes and sepsis lead to shedding of EG (*Martin et al., 2016*).

In the present study, the ventilator regime of LTV *versus* HTV had no relevant effect on the amount of EG shedding over the 6 h of the experiment. Considering that protective ventilation also requires more time to show positive effects in comparison to a high tidal ventilation regime, it is conceivable that the corresponding effects on EG could also be seen later. Also it must be noted, that the pathophysiology of ARDS is not only explained by shedding of the EG but also the alveolar epithelial barrier. A recent study showed elevated levels of heparan sulfate,a consituent of the alveolar epithelial glycocalyx and EG, in LPS-induced ARDS, which was not collected in our study. (*Li et al., 2021*). Maybe this parameter would have showed a difference between our experimental groups.

Finally, it is conceivable that the precision of the ELISA used was not sufficient to detect minor changes in the plasma HA levels that could indicate the differential effects of LTV and HTV on EG. The smaller but also significant increase in plasma HA concentration in the no LPS group in comparison to the LTV and HTV groups could be explained by the animal preparation. Furthermore, general anesthesia and mechanical ventilation have been found to lead to an impairment of pulmonary integrity, which could be seen especially in the lung injury score of animals that were mechanically ventilated (*Kamuf et al., 2018*; *Kamuf et al., 2020*).

Interestingly, data published by *Choi et al. (2006)* suggest that mechanical ventilation with higher tidal volumes and without PEEP during major abdominal surgery leads to increased plasma leakage and disruption of the endothelial barrier of the pulmonary vasculature compared with ventilation using lower tidal volumes and PEEP. If this observation is a consequence of EG depletion, one would assume that a difference in EG levels had to be detected at least when comparing the animals in the HTV and LTV groups. It can only be assumed that the expected effect was masked by septic shock. However, *Murphy et al. (2017)* observed that non-pulmonary sepsis resulted in greater degradation of EG than pulmonary sepsis. This may also suggest a superimposed effect of the septic shock model on subtle differences in EG degradation between the HTV and LTV groups that might otherwise have been detected.

The main limitation of our study is the missing direct measurement of EG. However, the indirect method of serum concentration of HA is widely accepted and has been used in several previous studies (*Dogné & Flamion, 2020*; *Smart et al., 2018*). Attempts were made to measure additional parameters, such as heparan sulfate and Syndecan-1, with commercially available ELISA kits. Unfortunately, no valid results could be measured with these kits in our model.

The septic shock model used for the LTV and HTV groups required the use of norepinephrine to ensure stable circulatory conditions. In this regard, MAP was maintained above 60 mmHg throughout the 6-h time period of the experiment. Higher MAP values might have been achievable with more generous fluid therapy. Considering potential damage to the EG from atrial natriuretic peptide release or volume overload, this option was discarded (*Jacob et al., 2013*).

The significantly higher dosage of norepinephrine in the LTV group in comparison to the HTV group can be explained by the significantly higher PEEP values in the LTV group; a correlation between the use of higher PEEP and lower MAP due to reduced preload caused by high intrathoracic pressures has been scientifically recognized (*Luecke & Pelosi, 2005*). The same mechanism serves as an explanation for the significantly reduced cardiac output in the LTV group after 6 h of the experiment. The blood count results showed a significant decrease in leukocytes and platelets in both the LTV and HTV groups; this phenomenon is known to be an immunological response to the administration of LPS and also a criterion for the diagnosis of sepsis. An interesting observation was made regarding hematocrit. It was significantly increased in both sepsis groups, which could be interpreted as a sign of hemoconcentration due to fluid loss across the endothelial barrier impaired by sepsis (*Rehm et al., 2004*).

Finally, the increased serum lactate levels in the LTV and HTV groups can be explained by impaired microcirculation resulting from septic shock. The small but significant decrease in the serum lactate levels in the no LPS group could be the result of slightly elevated levels at baseline after animal preparation. Neither the pulmonary wet-to-dry ratios nor the cerebral wet-to-dry ratios showed significant differences among the three experimental groups. The chosen time period of 6 h was based on experience from previous animal studies in which LPS was used to induce septic shock, although we used higher concentration of LPS to achieve ARDS based on experiences from preliminary prepared animals (*Lipcsey et al., 2010*). However, there is a possibility that a longer study period would be required to analyze a capillary leak in order to observe pulmonary edema and, more importantly, cerebral edema In retrospect, the choice of hydroxyethyl starch for hemodynamic stabilization proved to be less than optimal because it has beneficial effects on glycocalyx (*Zhao et al., 2020*). Additionally, hydroxyethyl starch is no longer used in sepsis due to its negative effects on various organ systems (*Rhodes et al., 2017*). This can be explained by the fact that, in the present study, this knowledge was not available at the time the experiments were conducted.

Although there were no significant differences in the lung injury score among the three groups, the parameters assessed generally showed high variances. However, it is important to note that the present study was not primarily designed or powered for this analysis. Moreover, several aspects need to be considered to further elucidate the results. A septic shock model with high doses of norepinephrine induces microcirculatory disturbances and, at the same time, LPS attenuates the mechanism of hypoxic pulmonary vasoconstriction (*Easley et al., 2009*). A sustained effect of pulmonary perfusion on the decrease in the oxygenation rate may explain these discrepancies. Additionally, the reported baseline oxygenation ratio values under baseline conditions were significantly

lower than data from recent studies from our laboratory with comparable animals (breed, age, weight, breeder). Therefore, a preexisting or contributory environmental influence cannot be excluded.

In conclusion, this study demonstrated that LPS-induced systemic sepsis caused EG degradation as measured by the plasma HA levels. In this porcine model, different ventilation regimes like LTV and HTV had no influence on EG degradation in the pulmonary or cerebral circulation. Further studies focusing on more EC-specific parameters and a longer study period are needed.

## ABBREVIATIONS

| | |
|---|---|
| **ARDS** | acute respiratory distress syndrome |
| **EG** | endothelial glycocalyx |
| **HA** | hyaluronic acid |
| **BL** | baseline |
| **LTV** | low tidal volume ventilation |
| **HTV** | high tidal volume ventilation |
| **$PaO_2\ FiO_2^{-1}$** | oxygenation ratio |
| **$V_t$** | tidal volume |
| **RR** | respiratory ratio |
| **PEEP** | positive end-expiratory pressure |
| **$FiO_2$** | fraction of inspired oxygen |
| **MAP** | mean arterial pressure |
| **PA** | mean pulmonary pressure |
| **CO** | cardiac output |
| **NA** | dose of norepinephrine |

### Funding
The study was funded by the German Research Council (DFG DA 842/2–2). The funders had no role in study design, data collection and analysis, decision to publish, or preparation of the manuscript.

### Grant Disclosures
The following grant information was disclosed by the authors:
German Research Council: DFG DA 842/2–2.

### Competing Interests
Erik K. Hartmann is an Academic Editor for PeerJ.

### Author Contributions
- Rainer Thomas conceived and designed the experiments, performed the experiments, analyzed the data, prepared figures and/or tables, and approved the final draft.

- Tanghua Liu conceived and designed the experiments, performed the experiments, prepared figures and/or tables, and approved the final draft.
- Arno Schad analyzed the data, authored or reviewed drafts of the paper, and approved the final draft.
- Robert Ruemmler analyzed the data, authored or reviewed drafts of the paper, and approved the final draft.
- Jens Kamuf analyzed the data, authored or reviewed drafts of the paper, and approved the final draft.
- René Rissel analyzed the data, authored or reviewed drafts of the paper, and approved the final draft.
- Thomas Ott analyzed the data, authored or reviewed drafts of the paper, and approved the final draft.
- Matthias David conceived and designed the experiments, prepared figures and/or tables, and approved the final draft.
- Erik K. Hartmann conceived and designed the experiments, analyzed the data, prepared figures and/or tables, and approved the final draft.
- Alexander Ziebart conceived and designed the experiments, performed the experiments, analyzed the data, prepared figures and/or tables, and approved the final draft.

## Animal Ethics

The following information was supplied relating to ethical approvals (*i.e.*, approving body and any reference numbers):

This study was conducted after approval by the State and Institutional Animal Care Committee (Landesuntersuchungsamt Rheinland-Pfalz, Mainzer Straße 112, 56068 Koblenz, Germany; Chairperson: Dr Silvia Eisch-Wolf; reference number: G12-1-059)

## Data Availability

The raw measurements are available in the Supplemental File.

## Supplemental Information

Supplemental information for this article can be found online at http://dx.doi.org/10.7717/peerj.12649#supplemental-information.

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
