# Peer review of "Hyaluronic acid plasma levels during high versus low tidal volume ventilation in a porcine sepsis model"

_PeerJ, doi:10.7717/peerj.12649_

## Round 0.1 · original submission · Major Revisions

Please address all the reviewer comments.

(additional) Major points:
1) Add a biomarker for glycocalyx injury such as sdc1.
2) Clarify number of animals (20 vs 26 in the manuscript!)
3) Clarify ventilation parameters

Reviewer 1 ·

Basic reporting

no comment

Experimental design

no comment

Validity of the findings

no comment

Additional comments

In general:
The experimental study by Thomas et al. aimed to investigate in a LPS induced porcine sepsis and ARDS model the effect of a lung protective ventilation (LTV) vs a non-protective ventilation (HTV) on endothelial glycocalyx shedding in the systemic, pulmonary and cerebral circulation at baseline (before) and after 3 hours and after 6 h of LPS administration. Parameters of mechanical ventilation, pulmonary gas exchange, hematology and hemodynamics were obtained throughout the study.
They found no differences in hyaluronic acid plasma concentrations (obtained from central venous blood) in the Intervention groups (LPS+LTV and LPS+HTV). They also state they found no differences in hyaluronic acid plasma concentration before and after brain and lung passage, but data is missing for the measurements of hyaluronan in the pulmonary and cerebral circulation (not shown in Figure 2).
The study on this clinically relevant field was performed thoroughly but is far from being mechanistic. The “negative” results are important findings to further assess the role of the endothelial glycocalyx in the pathophysiology of ventilator induced lung injury. Yet I see major weaknesses.
The four major weaknesses I see in this study are:
1. The manuscript does not make clear how glycocalyx shedding in sepsis is related to glycocalyx shedding in ARDS. Clarification of this detail is essential for the presentation of this study.
2. Clarity concerning experimental setup. I suggest to make the following more clear throughout the manuscript: Primary Intervention is LPS induced sepsis; this sheds EG as measured by hyaluronan plasma concentrations after 3 and 6 hours in this porcine model. A second hit by HTV does not seem to lead to further shedding of EG. To explore the protective effects of LTV a later time of sacrificed might be more accurate. The choice of model should be discussed in more clarity.
3. Conclusions from the findings in this study are not well depicted. Clearer Conclusions from the findings are indispensable and would contribute to the scientific quality of this study.
4. The lack of confirmatory parameters for endothelial glycocalyx shedding. The authors investigated only a single indirect parameter of glycocalyx shedding, the plasma concentration of hyaluronan. They attempted to test heparan sulfate plasma concentrations, but due to technical problems could not provide valid results. I suggest the authors test another parameter to support their data e.g.: Syndecan 1, or MMP9 (Dogné S, Flamion B. Endothelial Glycocalyx Impairment in Disease: Focus on Hyaluronan Shedding. Am J Pathol. 2020 Apr;190(4):768-780. doi: 10.1016/j.ajpath.2019.11.016. Epub 2020 Feb 6. PMID: 32035885.assessment.)
In detail:
The abstract is clear and concise; one remark:
49: sham ventilation might be confusing and the reader could assume another form of LTV. I suggest to name it “no LPS infusion”.
Methods clear and concise;
147: where can the reader find the defined schedule how the ventilator settings were adapted how did they chose PEEP settings? Please provide the used protocol.
Background: a small abstract to shedding of HA in brain and edema would be helpful to put the experimental setup for cerebral measurements and data into context.
Results: Figure 2 lacks the data which is stated in the text (50-51) pre- and post- pulmonary and brain Hyaluronan plasma concentrations.
Discussion
268-274: I would recommend to rewrite this paragraph to make it clearer to the reader that in the study of Coi et al. the mechanisms of action (atelect-trauma ZEEP plus abdominal surgery) differs from that in the underlying study (HTV vs LTV here with PEEP in LPS induced sepsis).
General comments and questions
Alveolar epithelial glycocalyx vs endothelial glycocalyx should be discussed in limitations (Li J, Qi Z, Li D, Huang X, Qi B, Feng J, Qu J, Wang X. Alveolar epithelial glycocalyx shedding aggravates the epithelial barrier and disrupts epithelial tight junctions in acute respiratory distress syndrome. Biomed Pharmacother. 2021 Jan;133:111026. doi: 10.1016/j.biopha.2020.111026. Epub 2020 Nov 24. PMID: 33378942; PMCID: PMC7685063.)
How where compliance data acquired? Please provide information.
Used much higher LPS concentrations than Lipscey et al.: 160 µg bolus then 16µg/kg/h continuously, as compared to 1µg/kg/h, what was the rationale for this difference? Please clarify.

Reviewer 2 ·

Basic reporting

1. The major concerns of this study are the unclear rationale and research question. As the authors stated, it has been reported that sepsis causes endothelial glycocalyx damage, which contributes to acute lung injury (Schmidt et al Nat Med 2012). While it is conceivable that ventilation could cause further damage to pulmonary vasculature after the establishment of ARDS, it is unclear how barotrauma leads to endothelial injury (and therefore, more HA shedding). I think that the manuscript can be strengthen by making a compelling argument in the introduction and providing more detailed background/context. I think this will lead to a clearly defined the study question. Perhaps authors can use references such as Choi et al 2006.

2. The article formatting is somewhat inconsistent throughout the manuscript. Half the sentences do not end with period which I think is the reference formatting issue. In line 248, references are formatted differently from the rest.

3. Minor misdefined abbreviation; it should be “hyaluronic acid”=HA, not “hyaluronic acid concentration”.

4. Heparan sulfate is included as a key word even though it was not measured in this study. It should be removed or replaced with hyaluronic acid to avoid confusion for readers.

5. In line 134, authors state that the animals that received LPS were randomly divided into 3 groups, while one of the groups did not receive LPS. This sentence needs to be edited.

Experimental design

1. In Line 142, it was stated that animals that did not develop ARDS were euthanized. I think that this set of animals could have served as a great control to demonstrate the effect of LPS-induced sepsis alone. HA levels in plasma can be mostly due to LPS-induced sepsis and this group would have demonstrated the contribution of ARDS to this shedding if any. The data from this group would have also clarified how the authors defined ARDS specifically and demonstrated (possible) difference in lung histology.

2. Although authors stated that 20 animals were bred for this study in method section, line 102, I believe it was actually 26, if the euthanized animals were accounted for.

3. Please provide number of biological replicates in figure legends.

4. Authors described their compliance to ethical guidelines with regards to animal transportation and euthanasia well.

5. I think that the method description for histological analysis could be improved. Although authors cited a study, it was unclear how authored scored the lung injury. It would be helpful for general readers if more description or examples of histological characteristics were shown. It was also unclear how many sections/field were evaluated per animal.

Validity of the findings

1. The most concerning point regarding the validity for the study findings is the lack of data that directly addresses the study question #2. Figure 2 shows HA ELISA data from central venous blood samples, but in the main text, it was described that authors collected samples from multiple catheter lines to determine HA levels in pulmonary and cerebral circulations. Was HA measured as described? One of the study questions, as stated in line 93-94 was to determine the extent of endothelial glycocalyx depletion in pulmonary and cerebral circulations for which plasma HA was supposed to be the readout. Lack of this data significantly diminishes the scientific merit of this publication.

2. Figure 3 can be improved by showing representative histological images.

3. Authors stated in the discussion (line 264-267) that mechanical ventilation itself can compromise pulmonary integrity. If that is the case, proper control (no mechanical ventilation) should be included to demonstrate the effect of mechanical ventilation to experimental parameters.

4. The conclusion statement (line 323-325) is not supported by the data provided. As mentioned above, the effect of different ventilation regimens to pulmonary or cerebral circulations was not measured specifically.

Additional comments

I understand that authors acknowledge the lack of direct evidence for EG degradation in their discussion, but I think that additional marker for EG would benefit the manuscript. If heparan sulfate ELISA did not work, perhaps syndecan-1 or chondroitin sulfate could serve as additional marker.

---

## Round 0.2 · Minor Revisions

Please reply to my major points from the previous review. Please reply to minor concerns by the reviewer of this revision.

Reviewer 2 ·

Basic reporting

My major concerns were addressed by authors in this revision.

Minor comments:
1. A reference appears to be missing on line 50, regarding brain edema.

Experimental design

Research questions were relevant and meaningful.
Ethical standard was withheld for the experiments.

Minor comments:
1. Authors responded and provided technical replicates for ELISA, rather than biological replicates. Please include biological replicates (number of animals) used for ELISA in figure legend for Figure 2.

Validity of the findings

Conclusion has been improved with the revision.

---

## Round 0.3 · accepted · Accept

Thank you for your revised version.